# Do Age-Friendly Community Policy Efforts Matter in China? An Analysis Based on Five-Year Developmental Plan for Population Aging

**DOI:** 10.3390/ijerph192013551

**Published:** 2022-10-19

**Authors:** Yongqiang Chu, Huan Zhang

**Affiliations:** 1Institute of Talent Assessment and Development for the Guangdong-Hongkong-Macau Greater Bay Area, Guangdong University of Finance & Economics, Guangzhou 510320, China; 2School of Social Development and Public Policy, Beijing Normal University, Beijing 100875, China

**Keywords:** age-friendly community, social participation, policy efforts, older adults, physical environment, China

## Abstract

(1) Background: The study will examine whether local governments’ policy efforts on age-friendly communities (AFC) promote older adults’ social participation in China. The extensive scope of AFC makes measuring policy efforts very challenging. The study attempts to introduce the developmental planning and goal-setting theory in public policy literature to answer this question. (2) Methods: We look at the Eleventh Five-Year Developmental Plan for Population Aging in subnational governments and CHARLS (the China Health and Retirement Longitudinal Study) baseline dataset from 2011, with data on policy strength and social participation of older adults. By using multilevel linear models, we regress social participation at the individual level on the policy strength of age-friendly communities at the provincial level. (3) Results: The results show that policy strength on AFC does vary substantially among provinces within China. And the interaction between policy strength of physical environment of local governments and community infrastructures is positively associated with social participation of rural older adults in China. (4) Conclusions: We conclude that policy efforts of local governments on the physical environment of age-friendly communities have effectively promoted the social participation of rural older adults in China. Policy makers could integrate physical infrastructures into their rural revitalization strategy to improve the wellbeing of Chinese older adults.

## 1. Introduction

Age-friendly community environments have been a hot topic in recent decades worldwide. Age-friendly environments are those in which ‘older adults are actively involved, valued, and supported with infrastructure and services that effectively accommodate their needs’ [1]. AFCs not only have positive effects on older adults’ health and wellbeing, but also promote aging in place, which can reduce the pressure on institutional care system [2,3]. In addition, AFCs are becoming popular because they can meet older adults’ personal preferences.

However, most AFC initiatives, while promising, are mainly being administered by non-governmental entities and supported by the private sector, which only have very limited success and minimal policy implications. Actually, the policy support from local governments is especially relevant in stimulating and enabling cities and communities to become increasingly age-friendly. Local government involvement has the strength to improve the community environment for older people on a larger scale and in a more sustainable way. Even though there is an emerging consensus that the policy support of local governments can greatly contribute to the development of AFC across the world [4,5,6], few studies have analyzed the specific effects and mechanisms of policy efforts on AFCs at a local level. Social participation is not only one of key characteristics of AFC [2], but also a key indicator of health and quality of life of older adults [7]. To bridge the research gap, this study will examine whether local governments’ policy efforts regarding AFCs significantly promote older adults’ social participation.

Social participation refers to a person’s involvement in activities that provide interaction with others in society or the community [8]. To older people, community-based activities, including volunteering, leisure pursuits, and physical activities, are the most popular. Age-friendly characteristics typically include opportunities to engage in meaningful activities, access to sources of social support, safe and pleasant neighborhoods and housing, transportation options beyond the personal automobile, and proximally located goods, services, and amenities [3,9]. Age-friendly environments usually include the social environment and the physical environment [2]. In most aging countries, governments actively construct an “opportunity structure” of social participation through the improvement of the physical and social environment which may provide essential conditions for older adults to participate meaningful social activities [10]. However, due to the wide scope of AFCs, the policies and measures adopted by local governments to promote the development of AFCs are generally extensive and fragmented. It is very hard to measure the policy efforts on AFCs at the local level in most countries. This is one of the important reasons why policy evaluation research in the AFC domain is very limited in the existing literature. The Chinese government has a relatively long-term commitment to an age-friendly environment and, since 2000, series of Five-Year Developmental Plans for Population Aging, a government-led, top-down approach, were adopted at multiple levels of government to mobilize multiple resources to invest in age-friendly cities or communities to promote aging in place. The promotion of social participation is also an important objective in this government-led plan. The case of China provides a valuable opportunity to examine the association between local governments’ policy efforts on the physical and social environment of AFC and social participation of older adults. The study tries to bridge the existing gap by exploring the effectiveness of China’s AFC governance framework featuring developmental planning.

## 2. The Context of Policy Efforts on Age-Friendly Communities in China

Developmental planning plays a leading role in China’s governance system, and the aging domain is no exception [11]. Recognizing the potentially salient influence of neighborhood physical and social contexts on older adults’ ability to age in place, the Chinese government has given more attention to the community contexts within which older adults reside. AFC initiatives are also integrated into developmental planning for population aging. From 2000 on, China established a Five-Year Developmental Plan for Population Aging, which includes the main content of the social and physical dimensions of AFC and includes specific targets linked to each dimension, broken down into five-year intervals to move toward a higher target. Specific government units have been assigned responsibility for different dimensions, including evaluating progress over time. Each local government needs to formulate and implement its own developmental plan, and its targets should not be lower than those of the higher government. The scope of AFC is extensive and fragmented, which inevitably involves multiple actors (different levels of government, sectors, and agencies) who should work in concert to achieve a set of objectives. To enhance the policy coordination of different departments, China established a developmental plan to guide local government’s investment into an age-friendly environment. As a coordinated system, China’s Five-Year Developmental Plan for Population Aging can direct activities of governments in formulating and implementing policies, which reflect the core intention and priority of corresponding governments. So far, China’s central and local governments at multiple levels have issued four Five-Year Development Plans for Population Aging. China has established an AFC governance framework featuring developmental planning. This gives us a valuable opportunity to evaluate the effectiveness of policy efforts on AFCs.

This study begins with a review of age-friendly communities in China, the role of government-led planning in China’s governance system for AFCs, and then pays special attention to the goal-setting theory in the public administration domain. The study aims to respond to whether the policy strength of AFCs of local governments promote older people’s social participation. We then propose the hypothesis concerning targets of AFCs in the Five-Year Developmental Plan for Population Aging and social participation of older adults. On this basis, by combing two datasets regarding the development planning objectives and the social participation of older adults in China, the study employs a multilevel linear model to demonstrate the effects of policy efforts on AFCs at the local level in China. Finally, we summarize the empirical results and highlight the policy implications for future research on age-friendly cities or communities.

## 3. Literature Review

### 3.1. Age-Friendly Communities in China

Age-friendly communities play an increasingly important role in China’s approach to population aging. In recent years, there has been an increasing number of studies on age-friendly communities in China. Some studies have found the characteristics of the AFC framework in the context of China. For example, Yi Wang, Gonzales, and Morrow-Howell used a nationally representative dataset in China to analyze community-level measures within the framework of AFCs and found that many AFC concepts did not apply well in developing areas [12]. Most articles have explored the health consequences of AFCs. Xie found that Chinese seniors’ perceptions of environmental age-friendliness were significantly associated with general life satisfaction and emphasized the potential role of age-friendly communities as having an influential force on older adults’ subjective well-being, regardless of their SES [13]. Li, Wang, and Nancy examine neighborhood effects on the health of Chinese older adults [14]. In recent years, some scholars have begun to explore the potential mechanism of the development of AFC in China from the perspective of public policy [15,16]. Chu emphasized that local governments can make valuable contributions toward making communities more age-friendly and found that the Chinese city government’s adoption of housing adaptation policy for older adults can be driven by local needs and then accelerated by interactions among neighboring governments [15]. Even though there is an emerging consensus that the policy support of local governments can greatly contribute to the development of AFCs in China, few studies have analyzed the specific effects and mechanisms of policy efforts on AFCs at the local level. This study tries to bridge the existing gap by exploring the effectiveness of China’s AFC governance framework featuring developmental planning. Specifically, the study focuses on whether the policy strength of AFCs of local governments promote older people’s social participation.

### 3.2. Planning as a Coordination Mechanism in China’s Governance System for AFC

The essence of planning is a governmental effort at policy coordination [11,17]. According to public management literature, ‘Coordination is the purposeful alignment of tasks and efforts of units or actors to achieve a defined goal’ [18]. It aims to improve the consistency of policy actions within the public sector, including three types of activities: information sharing, resource sharing, and joint action [19]. Considering the intergovernmental relations in China, we will introduce its coordinating process in policy areas for AFCs from two dimensions, vertical and horizontal [20,21].

Vertically, China’s planning process coordinates the policy priorities of central and local governments. In China, planning is a rational process of policymaking in the public sector. Generally, this process comprises of the following stages: agenda-setting, implementation, evaluation, and termination [22,23]. Firstly, China’s local governments can affect the policy priorities of the central government in agenda-setting stage. Agenda refers to “the list of subjects or problems to which governmental officials, and people outside the government closely associated with those officials, are paying some serious attention at any given time” [24]. Its core task is problem recognition and issue selection, which are inherently political processes in where political attention is attached to a set of relevant policy issues [22]. Actors within and outside government, such as local government officials, experts, and representatives of NGOs, constantly seek to influence and shape the agenda. Through the information sharing mechanism within the bureaucracy system, policy priorities set by local governments can be communicated to planners or policymakers in central government and may influence the problem recognition and issue selection of the central government. Secondly, in the process of implementation, the key question is, how central governmental bodies control subordinate units or use power. In most countries, the central government creates a set of institutions to monitor and supervise local governments to comply with national priorities [25]. Local governments who adopt the priorities of the central government set in the developmental plan will receive more financial incentives and legitimacy [11]. Moreover, government executives in local governments who comply with the plan will also be prioritized for promotion [11,26]. Then, the vertical consistency of policy actions at different levels of government can be achieved within the process of planning implementation in the shadow of hierarchy.

One core responsibility of this coordinating mechanism is to support the implementation of a series of recommendations from the Five-Year Developmental Plan for Population Aging in China. Firstly, the Commissions on Population Aging at different bureaucratic levels were authorized by the corresponding governments to guide, supervise, and inspect the lower-level government’s performance on implementing the plan [27]. In this way, vertical coordination of policies on AFC across different authority levels within the line of the Aging Commission is enhanced.

Horizontally, policy actions of multiple sectors at the same level can be properly organized to achieve the desired outcomes efficiently through planning [21]. In a situation where the intent is to pursue maximization of individual sector interest, there is a possibility that the outcomes will not be the achievement of the desired goals. A formal plan sets out a common purpose, prescribes solutions, and provides a basic policy framework for all sectors involved [17]. It structures implementation by delegating responsibilities and anticipated outcomes to related sectors so that they can input the right amount of resources at the correct time. These sectors then formulate their own, more detailed, operational, departmental policies, instructions, and other supportive policies to achieve better plan implementation [11]. In this operational framework, relevant sectors’ policy actions can be supportive to the implementation of developmental plans. The horizontal consistency between plan documents and specific policy actions of related sectors can be achieved based on this coordination system. Moreover, after the developmental plan was approved by the top authority at the start of the planning period, the member agencies started issuing a flurry of departmental policy documents such as decisions, opinions, programs, methods, or detailed rules to support the execution of the plan, especially plan agendas with quantified targets. Thus, the policy goals of member agencies can be consistent with plan goals at the same level of government. Given the above, developmental plans are a good proxy for government actions on age-friendly cities or communities.

### 3.3. Goal Setting and Public Service Performance

Goal setting is a central element of the planning system implemented by many nations [28]. A core argument is that the existence of accurate performance targets should lead to higher achievements. The publication of precise targets should strengthen managerial accountability to the public interest [29]. Supporters argue that, relative to vague aspirations made in planning documents, quantified targets help organizational members to focus on outcomes rather than processes provide a sense of direction, inspire them, and motivate them to work toward improved performance [30,31]. While there is controversy over the theory, most of the available evidence suggests that setting a quantified target is positively related to organizational performance, either directly or indirectly [32]. For example, Boyne and Chen (2007) tested this relationship based on objective judgements [28]. They used panel data for 147 English local education authorities between 1998 and 2003 and found that the extent of performance improvement was influenced positively by the presence of a clear target. These existing studies suggest the positive effects of goal setting in public management.

Specific to the AFC domain, goal setting in the Five-Year Plan for Population Aging also reflects the degree of the government’s commitment to promoting AFCs. Developmental goals without quantified targets are usually age-friendly cities or community declarations. On the contrary, developmental goals with quantified targets mean that governments will mobilize resources or adopt policies to make more substantive changes.

Thus, it is rational to use planning goals relating to AFCs in China’s Five-Year Developmental Plan for Population Aging as a proxy for policy efforts on AFCs of local governments in China. The purpose of this study is to demonstrate the effects and mechanisms of the AFC policy on the social participation of older adults. We formulate hypotheses regarding policy strength on AFCs at local level in China. Specifically, planning goals with quantified targets reflect higher emphasis and investment on AFCs of local governments than those without quantified targets. So, the study develops the following hypothesis: that older adults who live in jurisdictions with higher policy strength (quantified targets) on AFCs are more likely to participate social activities than their counterparts who live in jurisdictions with lower policy strength (non-quantified targets) on AFCs.

## 4. Research Methods

### 4.1. Analytical Strategy

We use a multilevel linear model to examine the association between policy strength on AFCs of local governments and social participation of older adults at the individual level, we develop a conceptual framework for analysis in this study (as shown in Figure 1).

Our framework is multilevel, with policy strength measures at the provincial level, physical environmental variables at the community level, and social participation at the individual level. A multilevel model is a perfect fit for the research questions that we are interested in [33]. Using multilevel models, we can deduce whether older people who live in provinces with quantified targets participate social activities more often, while taking into consideration community environment, demographic characteristics, SES, and health status at the individual level.

Our framework is longitudinal in nature, with policy measures at the provincial level being measured at one point in time and social participation measured five years later at the individual level. The Commission on Aging released its Five-Year Plan for Population Aging at the starting year of the plan period, and implemented two assessments, at the middle and terminal years of the plan period, respectively [11]. Taking the 11th Five-Year Developmental Plan for Population Aging as an example, which was formulated in 2006 and evaluated in 2008 and 2010, respectively. Thus, a five-year period is in accord with the plan period which is perfect to estimate associations between policy strength on AFCs and the subsequent social participation of older adults.

Is social participation of older adults associated with policy strength on AFCs? To this aim, we estimated a series of multilevel linear models [33]. The models accounted for a three-level hierarchical structure of the data, with individuals (level 1) nested within communities (level 2), and then within provincial-level jurisdiction (level 3). The basic equation for this model is shown below. Subscripts *i*, *j*, and *k* index, respectively, levels 1, 2, and 3.
$$Y_{ijk}=\beta_{0}+\beta_{1}x_{ijk}+\beta_{2}v_{jk}+\beta_{3}w_{k}+\gamma_{k}+\mu_{jk}+\varepsilon_{ijk}$$

In the model,$\;Y_{ijk}$ is the continuous outcome variable for the ith person in jth community, and in the kth province. $x_{ijk}$, $v_{jk}$, $w_{k}$ refer to explanatory variables, respectively, at levels 1, 2, and 3. The random intercept’s fixed component $\beta_{0}$ and the slopes $\beta_{1}$,$\;\beta_{2}$,$\;\beta_{3}$ are the parameters of the equation. $\varepsilon_{ijk}$, $\mu_{jk}$ and $\gamma_{k}$ refer to the variation in levels 1, 2, and 3, respectively, which are assumed to be normally distributed with constant variances of $\sigma_{\varepsilon }^{2}$, $\sigma_{\mu }^{2}$ and $\sigma_{\gamma }^{2}$. They refer to the unobserved variation in social participation at the provincial, community, and individual level, independent of variables in the model.

We gradually add the individual, community, and provincial-level variables and assess their associations with outcome variables. Model 1 is a three-level null model of individuals (level 1) nested within community (level 2) and then provincial jurisdictions (level 3), with no predictor variables in the model. There are only constant and error terms in levels 2 and 3 on the right side of the formula. This model provides a baseline for comparing the size of province-level variations in outcome variables in subsequent models. Model 2 includes all the individual and community level variables to assess how predictors at level 1 and 2 contribute to cross-province variations in outcome variables. Lastly, Models 3a and 3b add policy strength measures, respectively, to evaluate their associations with individual outcome variables. Models for continuous outcome variables are estimated with mixed in Stata 13.

### 4.2. Data

The analytic data set compiled for this study was derived from two sources: the 11th Five-Year Developmental Plan documents from provincial jurisdictions and baseline dataset of the Chinese Health and Retirement Longitudinal Survey collected in 2011.

Policy-level dataset. One significant feature of the study is that we use the Five-Year Developmental Plan for Population Aging as one data source to measure the policy strength on AFCs at the local level. Following the discussion on the planning-based coordination system in China’s bureaucracy system and goal-setting theory in the literature review, it is rational to use goal planning as a proxy for the involvement extent of the local government in promoting older adults’ social participation.

Community and individual-level dataset. The CHARLS is a nationally representative sample of over ten thousand older adults. The extensive size and geographic detail of the survey makes it possible to isolate the contribution of individuals and any disparities between provincial characteristics, while still accounting for local area variation in social participation. In addition, it also provides extensive standardized information on the socioeconomic status, health, and daily living arrangements of older people and also the characteristics of urban and rural communities. In the 2011 CHARLS baseline data, a total of 8075 individuals aged 60 and above participated in the survey, of which, 4849 respondents registered in a rural community and 3226 in an urban community. In the study, the units of analysis are the province-level jurisdiction and the individual-level older person.

The data structure required for this analysis is complex. We faced some constraints on our data selection. The Developmental Plan for Population Aging used covers a five-year period. The plan will be released in the first year of the five-year period and evaluated in the last year. Up to now, China has issued a total of five “Five-Year Development Plan for Population Aging”, including the 10th plan (released in 2001), the 11th plan (released in 2006), the 12th plan (released in 2011), the 13th plan (released in 2017), and the 14th plan (released in 2021). Accordingly, for the outcome variables, we can only select the following years of data, these are 2005, 2010, 2015, 2020, and 2025. Currently, only the CHARLS 2011 dataset both has nationally representative data and is closest to the time requirements of this study. The CHARLS 2018 could not meet the time requirements of the study, although these data have also been supplemented. The CHARLS 2011, while somewhat out of date, still supports our research question.

### 4.3. Measurement

Social participation. The study evaluated social participation by asking the respondents about the meaningful community activities they have undertaken in the last month. The activities this study pays attention to mainly refer to volunteering, leisure pursuits, and physical activities. For example, played Ma-Jong, played chess, played cards, or went to a community club; went to a sport, social, or other kind of club; took part in a community-related organization; did voluntary or charity work; attended an educational or training course. The respondents were asked about the frequency of the above activities (almost daily, almost every week, not regularly). The outcome is the sum of the frequencies of each of the above activities.

Policy strength on AFCs. China’s government tries to promote older adults’ social participation through the modification of the physical environment and the improvement of the social environment at the community level. The study measures policy strength on AFCs of local governments in these two domains to reflect policy efforts of different local governments on AFC. Table 1 lists the detailed targets of the developmental goals of AFC in social and physical dimensions, respectively. Policy strength on an age-friendly social environment is coded as a scale variable with values ranging from 0 to 9.

We ranked the provincial governments according to the specific value of a developmental goal from the highest to the lowest. The lowest value is assigned a value of 1, the slightly higher value is assigned a value of 2, and so on up to the highest value. Provinces that did not set numerical targets were assigned a value of 0. Then, we sum these scores (D1–D3) to get the policy strength on age-friendly social environment of local governments. The measurement of policy strength on age-friendly physical environments uses the same coding method. Policy strength on age-friendly physical environments is also coded as a scale variable with values ranging from 0 to 13. We also sum these scores (I1–I4) to get the policy strength on the age-friendly physical environment of local governments. Details of coding results for these two kinds of policies for provinces in China are given in Table 1.

Covariates were selected as potential confounding variables owing to the differential opportunities for participating in social activities [34,35]. The covariates are selected for our analysis based on the assumption of resource dependency [36], including demographic, socioeconomic, health, and community characteristics. Relevant demographic characteristics are gender, age, and partnership status (living with or without a partner). Then, we focus on educational attainment (the highest level of education completed) and log transformation of annual equivalized household income (standardized by the household size to get the household income, per standard person; household income includes gross incomes of all household members from employment, pension, private transfers, and capital asset income) as the main indicator of socioeconomic resources (SES). In the circumstances present in China, employment (paid work or not), pensioner (yes or no), and care-giving responsibilities (total time spent providing care to children, parents, and parents-in-law) are also significant to older adults. Furthermore, we include four interval variables for health status: self-report general health (1–6), mental health (1–40), ADL (0–18), and cognitive function (0–30). The higher the score, the worse the health status.

In addition, we also control the community environment, which maybe an important mediating mechanism to be tested in this study. The community environment mainly includes community physical infrastructures (0–7) (basketball court, swimming pool, outside exercising facilities, table tennis, room for card games and chess games, room for ping pong, and activity center for older people), and community organizations (0–5) (association for calligraphy and painting, dancing team or other exercise organizations, organizations for helping older adults, and older adults’ associations). All the above variables are treated as scale variables.

## 5. Results

Table 2 provides descriptive statistics for the individual-level variables, community-level variables, and province variables. The final study population was comprised of 4849 rural older adults and 3226 urban older adults distributed in provinces. Among older adults, the social participation score equals 0.29, lower than the social participation score of urban older adults (0.81). The demographic characteristics of older adults in rural and urban areas are very similar but the socioeconomic characteristics are quite different. For example, 64 percent of rural older people engage in paid work, but this percentage is only 30 percent in urban older adults. The percentage of older adults who have pensions in rural areas (18 precent) is also lower than that in urban areas (52 percent). The mean value of annual equivalized household income in rural area is 57,861 yuan (Chinese currency, RMB), almost 9184 U.S. dollars, and that of the urban families is 112, 278 yuan, almost 17,822 U.S. dollars. The health condition of rural older adults is slightly weaker than urban older adults. At the community level, the physical infrastructure index and social organization index of the rural community (1.37; 0.48) is much lower than the urban community (3.72; 1.87). At the province level, across the provinces in our study, the mean value of policy strength on the age-friendly physical environment and social environment is 1.25 and 3.32, respectively.

Figure 2 provides the pattern of bivariate associations between social participation—aggregated at the province level—and the macro-level variables ‘policy strength on social environment’ and ‘policy strength on physical environment’. It seems that the pattern of bivariate association between policy strength on the physical environment and social participation of rural older adults, and policy strength on the social environment and social participation of urban older adults, tends to support the hypothesis proposed above. A higher level of policy strength on the physical environment and social environment (on the *X*-axis) is generally paralleled by higher social participation. These figures serve as an initial exploratory investigation of whether policy strength variables may be associated with the social participation of older adults before examining the more complex multilevel models. The results of bivariate analysis showed that there is only a certain correlation between the social participation of rural older adults and policy strength on the physical environment of AFCs. Therefore, in the following, the article only presents the results of the multivariate analysis of the rural older adults in the results section. The study provides the results of the multivariate analysis of the urban older adults in the Appendix A section (Appendix A
Table A1).

Table 3 contains the results of the multilevel linear models for social participation of older adults in rural areas. The independent variables were included stepwise into the regressions. The first model for each group was a baseline model containing only the constant and level 2 and level 3 error terms (Model 1), control variables at individual and community level are then introduced in Model 2, while the two measures of policy strength on AFCs at province level are included in the final Models 3. At last, the two interaction terms between policy strength and community resources are successively included in Model 4a and 4b.

Table 3 indicates that social participation of older adults in rural areas varies across provinces (Model 1). The variance between provincial-level jurisdictions (level 3) is 0.0173, with a standard error of 0.008. The introduction of the individual and community-level variables in Model 2 reduce the variance component to 0.0112 and 0.0100 in Model 4b. Still, the estimated random effects remain statistically significant.

Our findings (Model 2) document the important role of individual and community level variables in determining the social participation of rural older adults. There is an evident gender gap, in that female rural older adults participate in fewer social activities. We detected a negative employment gradient in rural older adults, in that rural older adults who do paid work had a lower possibility of being socially productive. Interestingly, we also detected a positive association between caregiving and social participation. Educational status has a significant association with social participation. Health variables, including ADL, and cognitive function, are negatively associated with social participation scores of rural older people. Relative to community-based organizations, community physical infrastructure plays a significantly positive factor.

Models 3 shows the separate effects of policy strength of social environment and physical environment of AFC on the social participation of rural older adults. Unfortunately, although there are positive correlations between the two policy variables and the social participation, these relationships are not statistically significant.

In Model 4a, the interaction term (policy strength on physical environment and community physical infrastructure) has a significant positive correlation with social participation of rural older adults. This indicates that policy strength on physical environment of AFC may be through improving community infrastructures to promote the social participation of rural older people. However, in Model 4b, the positive correlation between the interaction term (policy strength on social environment and community-based organization) and social participation is not statistically significant.

## 6. Discussion

The study aims to evaluate the effectiveness of policy efforts on AFCs at the local level in China. Existing studies mainly focus on the health outcomes of age-friendly cities and communities [2,3], and why local governments adopt AFC initiatives [15,16,37], little research has been performed to evaluate the effectiveness of policy efforts on AFCs. One important reason is that the extensive scope of AFCs makes measuring policy efforts on AFCs very challenging. This study attempts to introduce targets in the developmental plan to reflect the policy strength on AFCs of different local governments and demonstrates the rationality of this approach.

Although previous studies have revealed the importance of the physical and social environment to age-friendly cities or communities [3,4,5], this study found that the policy strength of these two dimensions really varied at the local level in China. Further, community resources may play important role in mediating the relationship between policy strength on the physical environment of AFCs and social participation of older adults in rural China. That is, when local governments set higher policy strength on physical environment of AFCs, older adults who live in communities with more physical infrastructures are more likely to participate social activities. In the study, policy efforts on the physical environment of AFCs refer to policies concerning community physical infrastructures, which are necessary conditions for participating in meaningful social activities. This indicates that infrastructure is also important to the social participation of China’s rural older adults and that building community physical infrastructures maybe an effective policy solution to promote rural older adults’ social participation. In 2018, China released a new national strategy, that is, the Rural Revitalization Strategy [38]. One goal of this national strategy was to improve the living environment in rural China. The study finds that policy efforts of local governments concerning community physical infrastructures may contribute to rural older adults’ social participation. Thus, we suggest that local governments should actively incorporate AFC initiatives into their revitalization strategies to improve the community physical environment of rural older adults and promote aging in place.

In addition, this study is an initial attempt to address this problem from the perspective of governance. Developmental plans are an important governance tool that many countries around the world use to promote the development of age-friendly communities [11,17]. Considering the complex governance system for population aging in China, the study argues that the Five-Year Developmental Plan for Population Aging is a rational proxy for policy efforts on AFCs. Clear and quantified targets for future achievements lead to higher performance. We argue that, if one province adopts quantified targets for the physical and social environments of age-friendly communities, this province will make concrete policy interventions, for example, financial investment, not just a policy declaration. If one province cannot reach its target, it will be punished by the government at a higher level. Therefore, it is reasonable to examine the association between policy efforts on AFCs of local governments and the social participation of older adults by introducing the goal-setting of developmental planning.

There are also some limitations in the causality of this study. We integrate a time lag into our research design, that is, the policy variable is five years earlier than the individual outcome variable. Moreover, we control important confounding variables reported in the existing literature and consider the random effect of the community level. This is due to the importance of the community environment to everyday life for older adults. However, it must also be acknowledged that the study could not include enough information at the baseline so the “stretch” of individual social participation cannot be analyzed directly. With this limitation in mind, causal inference must be cautious and new analytical data must be developed in future studies.

## 7. Conclusions

Policy efforts of local governments on the physical environment of age-friendly communities have effectively promoted the social participation of rural older adults in China. Community infrastructure improvement should be a policy priority in this domain. At present, China is on the road to promoting a rural revitalization national strategy and policy makers should actively integrate physical infrastructures for older adults into their rural community development plans.

## Figures and Tables

**Figure 1 ijerph-19-13551-f001:**
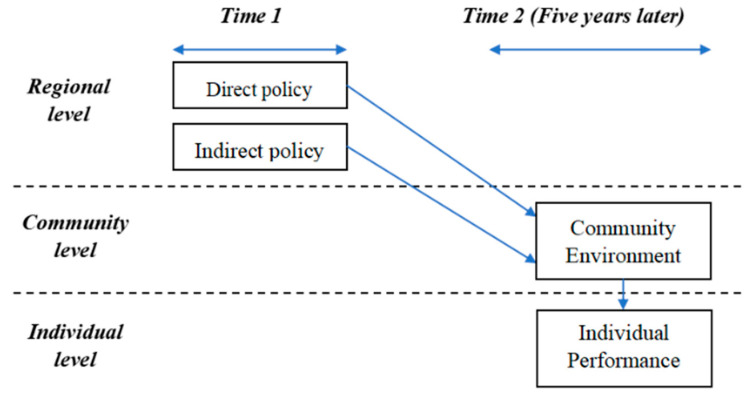
Provincial-level policy interventions to predict individual performance over time.

**Figure 2 ijerph-19-13551-f002:**
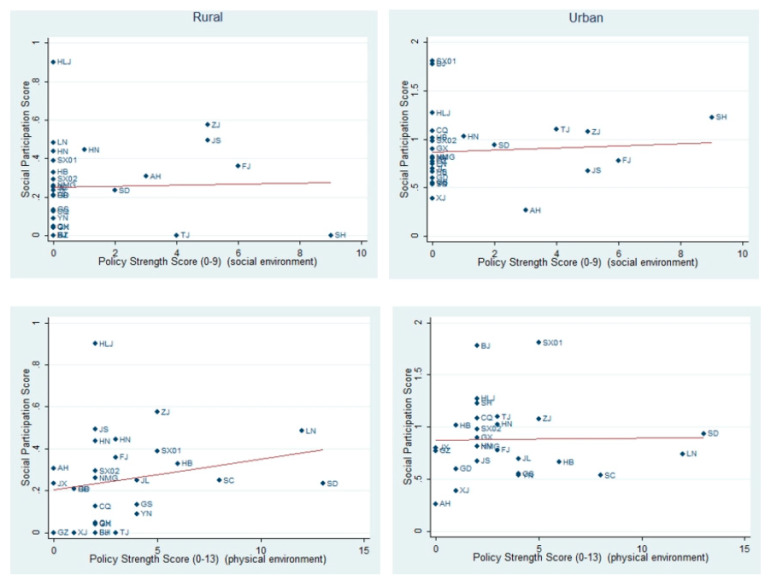
Bivariate Associations between Social Participation Score of older adults and Policy Strength Score in Provinces. Source: CHARLES 2011, Data coding from plan documents by authors.

**Table 1 ijerph-19-13551-t001:** Policy strength on AFC in China’s Five-Year Developmental Plan for Population Aging.

Category	Indicator	Label	Range
policy strength (Social environment)	The rate of older adults who participate cultural and sport activities regularly	D1	
	The rate of older adults who enrolled in community college for senior citizens	D2	0–9
	The rate of older adults who register as a volunteer	D3	
policy strength (Physical environment)	The coverage or number of physical spaces for older adults in the community	I1	
	The coverage or number of educational facilities for older adults in the community	I2	0–13
	The coverage or number of sport facilities for older adults in the community	I3	
	The coverage or number of cultural facilities for older adults in the community	I4	

**Table 2 ijerph-19-13551-t002:** Descriptive Sample statistics (unweighted).

Variables	Rural (*n* = 4849)	Urban (*n* = 3226)
	Mean (sd)	Mean (sd)
Social participation	0.29 (0.80)	0.81 (1.45)
Female	0.49 (0.50)	0.52 (0.50)
Age	68.41 (7.10)	68.70 (7.30)
Living with partner	0.73 (0.44)	0.76 (0.43)
Employment	0.64 (0.01)	0.30 (0.01)
Caregiving	3.06 (17.58)	3.08 (13.82)
Pensioner	0.18 (0.39)	0.52 (0.50)
Education level	2.39 (1.58)	3.52 (2.15)
Household income per person	57,860.79 (357,895.20)	112,277.80 (543,304.30)
Self-report health	4.04 (0.83)	3.90 (0.81)
Mental health	19.69 (6.45)	17.83 (5.81)
Activities of Daily Living (ADL)	1.18 (2.30)	1.11 (2.39)
Cognitive function	19.46 (5.93)	17.35 (5.61)
Community physical infrastructure	1.37 (1.74)	3.72 (2.42)
Community organization	0.48 (0.78)	1.87 (1.39
Policy strength (Physical environment)	3.32 (3.17)	
Policy strength (Social environment)	1.25 (2.38)	

**Table 3 ijerph-19-13551-t003:** Results of Multilevel Linear Models for the Social Participation Score of Rural Older Adults in China.

	Model 1	Model 2	Model 3	Model 4a	Model 4b
	b (se)	b (se)	b (se)	b (se)	b (se)
Female		−0.1086 *** (0.0246)	−0.1087 *** (0.0246)	−0.1084 *** (0.0246)	−0.1086 *** (0.0246)
Age		−0.0008 (0.0018)	−0.0008 (0.0018)	−0.0009 (0.0018)	−0.0008 (0.0018)
Living with partner		−0.0047 (0.0265)	−0.0051 (0.0265)	−0.0055 (0.0265)	−0.0057 (0.0265)
Employment		−0.0674 * (0.0266)	−0.0673 * (0.0266)	−0.0677 * (0.0266)	−0.0661 * (0.0266)
Caregiving		0.0016 * (0.0006)	0.0016 * (0.0006)	0.0016 * (0.0006)	0.0016 * (0.0006)
Pensioner		0.0740 * (0.0323)	0.0735 * (0.0323)	0.0731 * (0.0322)	0.0760 * (0.0323)
Educational level		0.0521 *** (0.0084)	0.0523 *** (0.0084)	0.0524 *** (0.0084)	0.0522 *** (0.0084)
Household income (log)		0.0074 (0.0043)	0.0070 (0.0043)	0.0068 (0.0043)	0.0070 (0.0043)
Self-report health		−0.0287 * (0.0141)	−0.0281 * (0.0141)	−0.0281 * (0.0141)	−0.0278 * (0.0141)
Mental health		−0.0032(0.0023)	−0.0032 ** (0.0023)	−0.0032 (0.0023)	−0.0031 (0.0023)
Activities of Daily Living (ADL)		−0.0143 ** (0.0051)	−0.0144 ** (0.0051)	−0.0144 ** (0.0051)	−0.0142 ** (0.0051)
Cognitive function		−0.0052 * (0.0020)	−0.0052 * (0.0020)	−0.0052 * (0.0020)	−0.0052 * (0.0020)
Community physical infrastructure		0.0543 *** (0.0118)	0.0527 *** (0.0120)	0.0340 * (0.0150)	0.0542 *** (0.0119)
Community organization		−0.0161 (0.0253)	−0.0193 (0.0253)	−0.0332 (0.0261)	0.0027 (0.0277)
Policy strength (Physical environment)			0.0083 (0.0069)	0.0028 (0.0072)	0.0074 (0.0069)
Policy strength (Social environment)			0.0137 (0.0149)	0. 0154 (0.145)	0.0352 (0.0187)
Policy strength (Physical environment) × Community physical infrastructure				0.0057* (0.0028)	
Policy strength (Social environment) × Community organization					−0.0205 (0.0109)
Random-effects: province	0.0173 (0.0080)	0.0112 (0.0058)	0.0100 (0.0051)	0.0091 (0.0048)	0.0100 (0.0051)
Units: province	25	25	25	25	25
Chi-Square	173.08	106.30	100.59	93.49	97.13
*p*-value	0.0000	0.0000	0.0000	0.0000	0.0000
*N*	4849	4849	4849	4849	4849

*** *p* ≤ 0.001, ** *p* ≤ 0.01, * *p* ≤ 0.05.

## Data Availability

The database was compiled with government register documents and national register representative datasets (CHARLS), researchers who are interested may contact the holders (Authority for Aging Population of Provincial Governments in China and Peking University) to obtain access to the registries used in this study.

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
