# Peer review of "Do Age-Friendly Community Policy Efforts Matter in China? An Analysis Based on Five-Year Developmental Plan for Population Aging"

_ijerph, 2022, doi:10.3390/ijerph192013551_

Round 1

Reviewer 1 Report

Please explain why did you select adults above 60.

Why you do not differentiate in your paper among the elderly by different age cohorts (e.g. 60-65; 65-70 ...) to determine variations among outcomes? 

Author Response

Thanks for this suggestion. It is interesting to determine variations in outcomes across age cohorts. The effect of age variable on the social participation of older adults is undoubtedly very important. In fact, the age variable has been included as a control variable in the model to further identify the possible effect of age. However, the purpose of the study is to examine the role of local governments in promoting AFC based on developmental planning for population aging, which actually does not set targets for different cohorts.

Reviewer 2 Report

What is the justification for only using CHARLS basline data from 2011?  These are extremely dated at this point.  It's my understanding that the data have been supplemented since then (wave 4 in 2018).  Even 2018 data is dated.  This must be addressed in any revision.

There are a number of papers that speak to the issues of AFC, and even in the China case. These are not represented here.  There a paper that uses that CHARLS data, too.  Literature review is not complete.

The subject of AFC is likely of interest to readers, but this particular approach, and given the methodological issues mentioned above, may give readers significant pause.

The paper itself uses acceptable analytical approaches, but my concerns are primarily with the data (recency) and grounding in existing literature and identifying precisely the research gap.

Author Response

Point 1: What is the justification for only using CHARLS baseline data from 2011?  These are extremely dated at this point. It's my understanding that the data have been supplemented since then (wave 4 in 2018).  Even 2018 data is dated.  This must be addressed in any revision.

Author response:

I would like to apologize for the misunderstanding. The reason for using CHARLS baseline data from 2011 is as follows:

First of all, the article is problem-driven research, which is dedicated to answer a relatively novel question in the domain of Age-Friendly Communities. The question is whether local governments’ policy efforts on Age-Friendly Communities promote older adults’ social participation in China. At present, the issue has not been adequately discussed in the existing literatures, and one of the important reasons comes from methodological and data challenges. The extensive scope of AFC makes measuring policy efforts very challenging. One of the potential contributions of the study is the introduction of the developmental planning and goal-setting theory in the public policy literature to respond to the challenge. Undoubtedly, the data structure required for this analysis are complex. We faced some constraints on our data selection. First of all, the developmental plan for population aging covers a five-year period. The plan be released in the first year of the five-year period and evaluated in the last year. Up to now, China has issued a total of five “Five-Year Development Plan for Population Aging”, including the 10th plan (to be released in 2000), the 11th plan (to be released in 2005), the 12th plan (to be released in 2010), the 13th plan (to be released in 2015), and the 14th plan (to be released in 2020). Accordingly, for the outcome variables, we can only select the following years of data, that is, 2005, 2010, 2015, 2020, 2025. Currently, only the CHARLS 2011 dataset both has nationally representative data and closest to the time requirements of this study. The CHARLS 2018 could not meet the time requirements of the study, although this data have been supplemented. Although the data is somewhat out of date, it still supports our research question.

Point 2: There are a number of papers that speak to the issues of AFC, and even in the China case. These are not represented here.  There a paper that uses that CHARLS data, too.  Literature review is not complete. The subject of AFC is likely of interest to readers, but this particular approach, and given the methodological issues mentioned above, may give readers significant pause. The paper itself uses acceptable analytical approaches, but my concerns are primarily with the data (recency) and grounding in existing literature and identifying precisely the research gap.

Author response:

Thank you for pointing this out. I have added a new section “Age-Friendly Communities in China” in the literature review. And the research method and measurement were explained in more detail.

Reviewer 3 Report

Dear Authors,

The article addresses an important topic related to the evaluation of the effectiveness of policies targeting seniors and their social activism. Unfortunately, in my opinion, the manuscript needs thorough revision. With the exception of the results and the discussion section, the study suggests that it is trying to answer the question posed in the paper without selecting the target group of older people understood as both urban and rural residents. Also, the hypothesis and purpose of the study do not suggest such a narrowing of the research problem. The results of the models are only for rural residents (Table 3). I believe that this is a big mistake in the whole study, as the specifics and conditions of life for people in rural and urban areas are different, also with regard to the formulation of development strategies of state authorities.  In addition, the very paragraph on lines 98-106 is not precise and does not inform what the research gap is, what question the authors pose or what hypotheses they have. In the case of Table 1 and the text referring to it, the units are not clear in the detail in terms of demographic characteristics.  The summary is laconic and does not refer to the results obtained. 

Author Response

Author response:

Thank you for pointing this out. I agree with reviewer’s idea on this point. That is the specifics and conditions of life for people in rural and urban areas are different, also with regard to the formulation of development strategies of state authorities. However, this study does not focus on "policy formulation and implementation", and only analyzes "objectives in the plan". In terms of the objectives, the Five-year developmental plan for population aging does not distinguish the difference between urban and rural areas. In fact, even though older adults in the urban area are very important, China's population aging acturally has a serious urban-rural inversion problem, with the rural older adults accounting for 70 percent of the total older population. Therefore, the realization of plan goals related to AFC undoubtedly needs to focus on the rural older population. This point is also reflected in this study.

Secondly, the study did not focus on rural older adults only. In fact, we also analyzed older adults in urban areas. However, because the association between policy variables and outcomes in the bivariate analysis were not significant, we did not take a further step to carry out the multiple regression analysis for the urban older adults. However, we agree with reviewer's idea that it is necessary to present the results for older adults in the urban area. Thus, we have made supplementary analysis for urban older adults (Table A1) and adjusted relevant content in the new revised draft according to the reviewer's suggestions. In addition, other issues pointed by the reviewer have also been revised in the manuscript. Thanks again for your advice.

Round 2

Reviewer 2 Report

The explanation is sufficient for point 1, but then it is not incorporated in any way that can discern into the paper itself.  So my question would remain (even if answered for me in peer-review - as a paper for public consumption - it would not be clear).  Point 2 is understood and the revision is appropriate.

Author Response

Point 1: The explanation is sufficient for point 1, but then it is not incorporated in any way that can discern into the paper itself.  So my question would remain (even if answered for me in peer-review - as a paper for public consumption - it would not be clear).  Point 2 is understood and the revision is appropriate.

Author response:

        Thanks again for your advice! I have added a new paragraph at the end of the Data section that explains the constraints we face in our data selection.

Reviewer 3 Report

Thank you for clarifying and adding additional results that are consistent with the article's assumptions. 

Author Response

Author response:

With regard to the English language and style, we have checked again and polished possible language problems.